# Adeno-Associated Viruses for Modeling Neurological Diseases in Animals: Achievements and Prospects

**DOI:** 10.3390/biomedicines10051140

**Published:** 2022-05-15

**Authors:** Evgenii Lunev, Anna Karan, Tatiana Egorova, Maryana Bardina

**Affiliations:** 1Institute of Gene Biology, Russian Academy of Sciences, 119334 Moscow, Russia; 2Marlin Biotech LLC, 354340 Sochi, Russia; akartar.n@gmail.com (A.K.); egorovatv@genebiology.ru (T.E.); 3Center for Precision Genome Editing and Genetic Technologies for Biomedicine, Institute of Gene Biology, Russian Academy of Sciences, 119334 Moscow, Russia

**Keywords:** genetic neurological disorders, animal models of human disease, viral vectors, adeno-associated viruses, transgene delivery

## Abstract

Adeno-associated virus (AAV) vectors have become an attractive tool for efficient gene transfer into animal tissues. Extensively studied as the vehicles for therapeutic constructs in gene therapy, AAVs are also applied for creating animal models of human genetic disorders. Neurological disorders are challenging to model in laboratory animals by transgenesis or genome editing, at least partially due to the embryonic lethality and the timing of the disease onset. Therefore, gene transfer with AAV vectors provides a more flexible option for simulating genetic neurological disorders. Indeed, the design of the AAV expression construct allows the reproduction of various disease-causing mutations, and also drives neuron-specific expression. The natural and newly created AAV serotypes combined with various delivery routes enable differentially targeting neuronal cell types and brain areas in vivo. Moreover, the same viral vector can be used to reproduce the main features of the disorder in mice, rats, and large laboratory animals such as non-human primates. The current review demonstrates the general principles for the development and use of AAVs in modeling neurological diseases. The latest achievements in AAV-mediated modeling of the common (e.g., Alzheimer’s disease, Parkinson’s disease, ataxias, etc.) and ultra-rare disorders affecting the central nervous system are described. The use of AAVs to create multiple animal models of neurological disorders opens opportunities for studying their mechanisms, understanding the main pathological features, and testing therapeutic approaches.

## 1. Introduction

Neurological disorders affect up to one billion people and remain the leading cause of long-term disability and one of the leading causes of death worldwide [1]. In neurological diseases, the functioning of the central nervous system (CNS) as well as peripheral nerves is impaired, leading to a wide range of clinical symptoms, including seizures, motor impairment, behavioral and cognitive disturbances, mental deterioration, and many others.

Advances in genetic diagnosis have revealed the underlying mutations for many neurological conditions. Disorders caused by mutations in a single gene are referred to as monogenic, e.g., Huntington’s disease and spinal muscular atrophy (SMA). Complex disorders, such as Alzheimer’s disease, Parkinson’s disease, and major depressive disorder are caused by a combination of genetic and environmental factors. Genetically determined neurological diseases also vary in severity and can manifest at different ages. Early-onset disorders can first become evident in infancy and take the form of neurodevelopmental delay, epileptic encephalopathy, movement disorders, etc. Neurodegenerative diseases usually manifest in adulthood and are characterized by progressive loss of neuronal subsets. The most common diseases in this group are Alzheimer’s disease, Parkinson’s disease, Huntington’s disease, ataxias, and amyotrophic lateral sclerosis (ALS). Neurological disorders also are grouped by the inheritance pattern (dominant or recessive) and the effect of the disease-causing mutation on the function of the gene product. Some mutations result in the protein or RNA product with enhanced or altered functional activity, which is toxic to the cells. An example is an abnormal expansion of the DNA repeats in the causative genes. Mutations with repeat expansion are typically dominant and cause so-called repeat expansion disorders, including Huntington’s disease, different spinocerebellar ataxias (SCAs), certain forms of ALS, and others. By contrast, mutations can also lead to the inactivation of the gene products. Neurological pathologies associated with the deficiency of the functional protein are most often recessive, and some examples include a group of lysosomal storage diseases [2] and SMA [3].

Animal models of human neurological diseases are an indispensable tool for studying pathogenesis and testing therapeutic approaches. Disease modeling is routinely done in laboratory mice and rats and less frequently in large animals such as rabbits, dogs, pigs, goats, and non-human primates (NHP). A straightforward way to model diseases is to introduce disease-causing mutations into the animal genome by means of transgenesis or genome editing [4]. Unfortunately, the wide use of these techniques is limited due to the laborious procedures, the long time required to create a single disease model, and the need for special equipment. Moreover, methods are tailored to specific animal species. Neurological disorders might be complicated to model by genome manipulation technologies due to the reduced survival of genetically modified animals with severe disease-causing mutations. Adult-onset of some neurological disorders also requires conditional expression control of the engineered allele. 

An alternative approach to disease modeling in animals is using viral vectors for transgene delivery. The advantage is that the same vector can be used as a delivery vehicle to introduce disease-causing transgene into various species, from mice to NHP. Moreover, viral vectors allow targeting a specific area of the brain and the temporal control of the transgene expression. The relative simplicity of the viral gene transfer and the ability to combine models without creating multiple transgenic lines is also a plus. Adenovirus, herpesvirus, and lentivirus vectors are all used in neuroscience studies for gene delivery [5]. However, recombinant adeno-associated virus (rAAV) vectors gained distinct popularity for modeling neurological disorders. Wild-type AAVs are small, non-pathogenic viruses with a simple structure convenient for genetic engineering. AAVs efficiently infect dividing and non-dividing cells; however, they are not competent to self-replicate, relying on co-infection with helper viruses (adenovirus, herpes virus) to complete their life cycle [6]. Natural AAV serotypes demonstrate broad cellular tropism infecting the CNS, cardiac and skeletal muscles, liver, kidney, lung, and other tissues. These properties make AAV an attractive tool for gene transfer applications, including basic neuroscience research [5,6,7] and the development of gene therapy [6]. The US Food and Drug Administration has already approved the first AAV-based drugs for patients with inherited disorders of the CNS: Zolgensma for SMA and Luxturna for Leber congenital amaurosis [8]. Notable advancements in preclinical and clinical research on AAV-based gene therapy for other CNS disorders have been comprehensively reviewed [9,10].

Here, we describe a lesser-known application of AAV vectors, namely as gene delivery vehicles for creating animal models of human neurological disorders with a genetic cause. The review focuses on studies that used AAVs either to deliver the disease-causing gene or to directly impact the expression level of the endogenous causative gene in the model animals. Designs of the AAV expression vectors, components that provide neuro-specific targeting, and routes of the AAV administration into the nervous system are discussed. The review presents the most striking examples of modeling some of the most common and rare disorders that primarily affect CNS functions (e.g., Alzheimer’s disease, Parkinson’s disease, Huntington’s disease, ALS, SCAs, epilepsies, and depression). Disease modeling utilizing AAV and genome manipulation enzymes (such as Cas9 endonuclease and Cre recombinase) or instruments for opto- and chemogenetics are beyond the scope of this review and were well described in the recent works [7,11,12]. 

### Recombinant Adeno-Associated Viruses (rAAVs)

Wild type AAV is a member of the genus Dependoparvovirus within the virus family Parvoviridae. A viral particle consists of a single-stranded DNA genome packaged into a non-enveloped icosahedral capsid (26 nm) (Figure 1A). Genomic DNA (4.7 kb) that encodes replication (*rep*) and capsid (*cap*) genes is flanked with hairpin-like inverted terminal repeats (ITRs, 145 bp) required for genome packaging, replication, and synthesis of the transcription competent double-stranded form. 

To model human neurological disorders in animals, researchers create rAAVs using genetic engineering. Protein-coding genes in the AAV genome are replaced with a custom expression cassette of up to 4.5 kb (Figure 1B) [13]. The only viral *cis-elements* left in the vector are ITRs required for packaging the expression cassette into the protein capsid of the chosen neurotropic serotype. Such a recombinant viral genome can be cloned in plasmid DNA and introduced into the producer cells. Capsid proteins and viral proteins necessary for AAV particle assembly and vector replication are supplied in trans in the producer cells [6]. rAAV production and purification methods are well established and allow obtaining virus preparations with a high titer and purity suitable for in vivo applications, including delivery into the CNS [14]. 

#### 1.1.1. AAV Expression Cassette

The elaborate design of the cassette placed between the ITRs in the AAV expression vector is crucial to successful disease modeling. Expression cassettes typically consist of a promoter, a *transgene*, and a transcription terminator (Figure 2) [15]. The transgene is the payload of the vector, and it varies greatly depending on the putative pathogenetic mechanism of the neurological disease.

The most common type of transgene is protein-coding sequences (CDS) that frequently carry a causative mutation (Figure 2A). These transgenes are applied for modeling dominant neurological diseases caused by producing the gene products with enhanced, altered, or new functional activity. Such disease-causing mutations are classified as *gain-of-function* (GoF) or *dominant-negative*. AAV-mediated expression of a malfunctioning protein or its fragment can be used to reproduce the pathogenic effects in wild-type animals. In particular, this approach is used to model repeat expansion diseases such as Huntington’s disease [16] and spinocerebellar ataxia type 6 [17] by including CDS with the pathogenic repeats in the AAV expression cassette. Another group of repeat expansion disorders is characterized by an increased number of repeats in the non-coding gene regions (5′ and 3′ untranslated regions, introns), leading to the production of the toxic RNA [18]. Modeling of such disorders can be exemplified by chromosome 9 open reading frame 72 (C9orf72)-related ALS and frontotemporal dementia (FTD) when repeat-containing non-coding sequences are used as transgenes and generated in the cells via AAV-mediated transcription [19,20]. Protein-coding transgenes can also be used to confirm or disprove the dominant mechanism of the clinical variant. For example, AAV-delivered transgenes with causative mutations were used to demonstrate the dominant-negative fashion of the mutation for some forms of epilepsy [21] and spinocerebellar ataxia [22]. Elevated or dysregulated expression levels of the wild-type protein or precursor can also cause the onset and progression of the neurological disease. Therefore, AAV vectors can be used to overexpress causative genes in the neurons to create pathogenic conditions in a healthy animal model. Examples of such studies are described below for some forms of ALS [23], Alzheimer’s disease [24], epilepsy [25,26], proBDNF [27], and SIRT1-dependent depression [28]. 

Fundamentally different types of transgenes are used for gene suppression to model neurological conditions caused by a protein deficiency (Figure 2B). These transgenes encode short RNA molecules such as small hairpin RNA (shRNA) and artificial microRNA (miRNA). Following AAV-mediated delivery and expression, shRNA and artificial miRNA silence specific endogenous genes by inducing RNA interference (RNAi) [29]. The shRNA and miRNA differ in the design of the hairpin structure [30]. The shRNA is classically transcribed as a sense-loop-antisense sequence of unpaired nucleotides. By contrast, miRNA have a more complex stem-loop structure and utilize a modified backbone of the natural pri-miRNA transcript (e.g., miR-30a, miR-155) subject to additional processing steps by cellular machinery. AAV-mediated gene suppression is used for modeling diseases that typically have a recessive inheritance pattern and occur in the case of *loss-of-function* mutations (LoF) in the gene. For instance, a shRNA-expressing AAV vector was used to mimic SMN deficiency in wild-type piglets and induced the development of SMA-like symptoms [31]. Gene suppression approach can also be used for modeling lysosomal storage diseases such as Gaucher disease, which has a subtype affecting the brain [32].

The promoter is another element of the expression cassette that requires careful selection and defines the efficiency and cell specificity of the transgene synthesis [7,15,33]. The expression of the protein-coding transgenes is controlled by the RNA polymerase II (Pol II) promoters (Figure 2A). Small natural promoters (EF1A, PGK1, GUSB) drive transgene expression in neuronal cells at a low level [15,34]. Constitutive viral promoters, such as the CMV promoter, provide high levels of transgene transcription; however, they are prone to silencing in neuronal cells [35]. Therefore, AAV vectors with chimeric promoters such as CBA and CBh are often used for the long-term transgene expression in the CNS [36]. Neuron-specific promoters such as hSyn and CaMKII allow to restrict transgene expression in non-neuronal tissues; the L7-6 promoter is highly specific to Purkinje cells and used for cerebellar research [37]. The GfaABC1D promoter (a truncated variant of GFAP) drives expression in astrocytes [7] and is utilized for modeling epilepsies [25,26]. Transcription of the small non-coding RNAs in AAV vectors is driven by ubiquitous RNA polymerase III (Pol III) promoters, e.g., U6 and H1 promoters (Figure 2B) [33]. However, several studies reported neuronal toxicity following AAV-delivered shRNA expression from strong Pol III promoters, which presumably led to the saturation of the endogenous RNAi machinery [38]. Synthesis of the miRNA transcripts can also be mediated by constitutive or tissue-specific Pol II promoters that provide a more physiological level of expression. 

To ensure sustained transgene synthesis in vivo, additional elements are included in the expression cassette regulating expression at the posttranscriptional level (Figure 2A) [15]. For instance, the intron in the 5′-UTR increases transcript stability, while the woodchuck hepatitis virus posttranscriptional regulatory element (WPRE) in the 3′-UTR enhances export to the cytoplasm [15]. For some modeling studies, expression of the reporter protein (EGFP, EYFP, Venus, tdTomato, etc.) along with the disease-causing transgene is desirable [21,39]. This is solved by creating AAV vectors with the bicistronic expression cassette [7]. 

While the customized design of the AAV expression vector is the first and the most critical step in disease modeling, delivery of the vector into neuronal tissues presents a separate challenge.

#### 1.1.2. Targeting Neuronal Tissues

To reproduce the pathogenic phenotype, transgene expression should localize in the tissues and the subtype of neurons that are primarily affected by the disease. This implicates that for some neurological disorders, AAV-mediated delivery should be directed to a specific brain region (e.g., cerebellum, striatum, substantia nigra, hippocampus), whereas for other conditions broader transgene expression in the CNS is required, including targeting of the cerebral cortex, spinal cord, motoneurons, etc. (Table 1). Desired biodistribution profile can be achieved by using AAVs of selected serotypes combined with the curated route of vector administration and optimal dose (Figure 3).

Tropism of the AAV serotypes to the CNS has been extensively studied in rodents and large lab animals [40,41]. The efficient transduction of the neuronal tissues in vivo has been shown for natural serotypes (1, 2, 5, 6, 8, 9, rh8, and rh10) and engineered AAV capsids such as ANC80L65, PHP.B, or PHP.eB, which is a variant of PHP.B with enhanced CNS infectivity, enabling the reduction of the viral load [7,8]. While most serotypes exhibit neuronal tropism, serotypes AAVDJ8 and ANC80L65 can infect astrocytes [40,42], and serotype Olig001 was generated to transduce oligodendrocytes [15]. Figure 3 and Table 1 list the most popular neurotropic serotypes of AAVs used by researchers for modeling neurological disorders in rodents and NHP. 

In addition to the serotype, AAV transduction of the central nervous system is determined by the route of virus administration into lab animals (Figure 3). The most common techniques include intraparenchymal, intracerebroventricular, and intravenous injections [8,42]. The choice of the AAV administration method for modeling is determined by the disease itself, as well as the availability of special equipment (Table 1). Intraparenchymal injections of AAV vectors are applied to target specific areas of the brain and result in the local transgene expression. Hence, intraparenchymal injections in adult animals are performed to induce neurodegenerative disorders for which the affected brain area has been defined. For instance, modeling of Parkinson’s disease is done by intraparenchymal delivery of AAVs into the substantia nigra [43,44], Huntington’s disease is typically simulated with intrastriatal injections [45,46,47,48], and Alzheimer’s disease can be induced by intrahippocampal injections [24,49,50]. Targeting of the cerebellum, in addition to several other brain zones, is achieved by intracerebroventricular injections and is used in SCA studies [17,22]. Intrathecal injections are particularly useful for modeling diseases that affect motor neurons, such as SMA [31] and ALS [19]. Global transgene expression in the brain can be achieved by systemic AAV delivery, especially when administered into neonates [51]. Indeed, several AAV serotypes (AAV8, AAV9, PHP.B) can penetrate the blood–brain barrier when injected into the tail vein or retro-orbital sinus and demonstrate a broad profile of AAV spreading in the CNS. This route of administration is advantageous for transducing the spinal cord and multiple regions of the brain, including the cerebral cortex [37,51]. The exotic case describes the intramuscular injection of AAV to simulate neurological pathology [27]. In this case, the role of the peripherally expressed neurotrophic factor as the depression trigger was addressed, and the skeletal muscle served as a biofactory for efficient transduction by AAV and a high level of transgene synthesis and secretion into the bloodstream.

To achieve a sufficient level of AAV transduction, the virus dose should be optimized (Figure 3). Typically, high titer AAV stocks (10^12^–10^13^ GC per mL) are required to target the CNS. The AAV dose used for the direct CNS injections on average is lower than the amount of the virus needed to target the brain by systemic delivery. Depending on the target CNS area, the doses used for local injections vary greatly and range from 10^8^ to 10^11^ GC for rodents and from 10^10^ to 10^12^ GC for NHP. Systemic delivery is often used on rodents with an average dose 10^13^ GC/kg (genome copies per kilogram of the body weight). Newborn mice are convenient objects for AAV-mediated disease modeling and require low virus loads as compared to adult rodents and NHP (Table 1). The administration of a high dose of AAVs can lead to neurotoxicity—for example, the development of neurodegenerative processes, neuroinflammation especially when injected into the CNS, or hepatotoxicity in the case of systemic administration. The toxic effects of AAVs were discussed in detail in many reviews of gene therapy preclinical and clinical studies [10,52]. 

Thus, efficient delivery of the transgene to the desired area of the CNS can be ensured by the selection of the AAV serotype, the route of the administration and the vector dose. Following transgene expression in the target tissues, the pathogenic phenotype can be observed 1–6 months after AAV injections depending on the disorder. In rapid-onset models, the symptoms can be detected as early as 2–3 weeks after AAV delivery. Interestingly, the researchers found that episomal expression of the AAV vector persists in neurons for at least 15 years after administration [53]. For modeling poorly studied neurological diseases and utilizing novel AAV vectors, it is recommended to conduct preliminary experiments to verify conditions making use of AAV with reporter genes [7,12]. Below, we provide examples of using AAV to model various CNS human disorders with the genetic component.
biomedicines-10-01140-t001_Table 1Table 1Key features of experiments on AAV-mediated targeting of different areas of the CNS to model neurological disorders.DiseaseThe Most Affected AreaAnimalsAge of Animals *rAAV SerotypeDose **Injection Route ***ReferenceAlzheimer’s diseaseHippocampus, the associative cortex and subcortical structuresMouseadultAAV11 × 10^9^ TU; 2 μLIP, hippocampus[50]MouseadultAAVrh105 × 10^8^–10^9^ GC/side; 2 μLIP, hippocampus[49]RatadultAAV13 × 10^10^ GC/side;3 μLIP, hippocampus[24]RatadultAAV93 × 10^9^ GC; 3 μLIP, substantia nigra[44]NHPadultAAV11.176 × 10^13^ GC/μL sample used, injection volume not specifiedIP, entorhinal cortex[54]Parkinson’s diseaseSubstantia nigraRatadultAAV62 × 10^10^ GC; 2 μLIP, vagus nerve[55]RatadultAAV62.5 × 10^10^ GC/side; 4 μLIP, substantia nigra pars compacta[43]Amyotrophic lateral sclerosis, frontotemporal dementiaMotor neurons, Frontal and temporal lobesMouseneonatalAAV94.5 × 10^10^ GC; 5 μLIT, cisterna magna[19]MouseneonatalAAV93 × 10^10^ GC/side;2 μLICV[20]Amyotrophic lateral sclerosisMotor neuronsRatneonataladultadultAAV9AAV-PHP.BAAV9,AAV-PHP.BAAV96.7 × 10^14^ GC/kg2.7 × 10^14^ GC/kg3 × 10^13^ GC/kg~1.4 × 10^10^ GCIVICV[51]RatadultAAV91–6 × 10^13^ GC/kgIV[56]RatadultAAV14.5 × 10^9^ GC;1.5 μLIP, C6 of the spinal cord[23]NHPadultAAV11.5 × 10^10^ GC or 1.5 × 10^11^ GC;5 μLIP, C5–6 of the spinal cord[23]Huntington’s diseaseStriatumMousejuvenileAAV-DJ1.24–2.46 × 10^9^ GC/side; 2 μLIP, striatum mid-coronal level[47]MouseadultAAV1/8 mosaic vector 3.15 or 5.4 × 10^10^ GC/side; 3 μLIP, striatum[45]MouseadultAAV53.45- 7 × 10^10^ GC/side; 0.5 μLIP, hypothalamus[57]MouseadultAAV62.4 × 10^9^ VP/side;2 μLIP, layer V of the cortex[16]RatneonatalAAV51.4 × 10^11^ GC/side; 1 μLIP, dorsal striatum[46]RatjuvenileAAV91.52 × 10^11^ GC/side; 4 μLIP, striatum[58]RatadultAAV24 × 10^9^ or 4 × 10^10^ GC; 2 μLIP, striatum[48]Ratadultchimeric AAV1/AAV23 × 10^9^ GC/side;3 μLIP, striatum[59]NHPadultAAV65.3 × 10^12^ VP;200 μL/4 sitesIP, striatum[16]KCNMA1-related cerebellar ataxia ****Cerebellum (Purkinje cells)MouseneonatalAAV92 × 10^12^ GC/side;2 μLICV[22]Spinocerebellar ataxia type 6Cerebellum (Purkinje cells), spinal cordMouseneonatalAAV91 × 10^10^ GC;2–4 μLICV[17]Spinocerebellar ataxia type 1 and 3MousejuvenileAAV-PHP.B and AAV-PHP.eB~6.5 × 10^13^ GC/kgIV[37]Spinal muscular atrophyAnterior horns of the spinal cordPigneonatalAAV96.5 × 10^12^ GC/kgIT, cisterna magna[31]Goucher disease type 3unknownMouseneonatalAAV12 × 10^10^ VP/side;2 μLICV[32]GNAO1-encephalopathyunknownMouseadultAAV93 × 10^9^ GC/side;300 nLIP, dorsal striatum[21]Focal neocortical epilepsiesNeocortexMouseadultAAV85 × 10^8^ GC; 0.5 μLIP, cortex[26]Refractory epilepsy Hippocampus and many other regionsMouseadultAAV82 × 10^9^ GC; 2 μLIP, CA3 region of the hippocampus[25]Major depressive disorderAmygdala, hippocampus and many other regionsMousejuvenileAAV95 × 10^8^ GC/side;0.5 μLIP, ventral hippocampal dentate gyrus[60]MouseadultAAV92.16 × 10^9^ GC; 7.9 × 10^8^ GC/side;0.2 μL IP, basolateral amygdala[28]MouseadultAAV81 × 10^11^ GC/side;25 μLIM, quadriceps[27]MouseadultAAV25 × 10^9^ GC/side;0.5 μLIP, basolateral amygdala[61]Ratadultnot specifiednot specifiedIP, dorsal hippocampal dentate gyrus[62]RatadultAAV91–1.5 × 10^9^ TU;1–1.5 μLIP, CA1 region of hippocampus[63]Ratnot specifiednot specified3 µL of virus, titer not specifiedIP, prefrontal cortex[64]* Rodents were divided into three groups according to age: Neonatal—from birth to one week, juvenile—from 1 to 8 weeks, adult—more than 8 weeks. All studies with NHP were conducted on adult animals. ** The titers of viral samples are indicated in TU—transfection units, GC—genome copies, and VP—viral particles. *** Injection sites are described as IP—intraparenchymal, IT—intrathecal, ICV—intracerebroventricular, IV-intravenous, and IM—intramuscular. **** The set of symptoms caused by the discussed mutation in the KCNMA1 gene does not yet have an official name as a disease, so the name indicated in the table is not officially recognized, but includes the name of the mutant gene and the main symptom. The authors of the article believe that in the future this disease will most likely become one of the types of SCA.


## 2. Examples of Modeling Neurological Diseases

### 2.1. Alzheimer’s Disease

Alzheimer’s disease (AD) is a multifactorial neurodegenerative disorder with cognitive decline that affects 1 in 10 people over the age of 65. Histopathological analysis reveals two features in the patients’ brains: accumulation of amyloid-β (Aβ) and hyperphosphorylated tau protein [65]. An autosomal dominant manifestation of the disease is associated with mutations in the *APP* (encoding amyloid precursor protein), *PSEN1* and *PSEN2* genes (encoding presenilin-1 and presenilin-2, respectively). These proteins are involved in the generation and processing of Aβ [66]. The hippocampus, as well as the associative cortex and subcortical structures, are most significantly affected in AD [67]. Despite significant advances in recent decades, researchers are still far from understanding the mechanisms of this disease and animal models are extremely useful for investigating the triggers of AD pathology. Rodents do not naturally develop the pathology of AD, making it difficult to model the disease [68]. To overcome the limitation in rodents, researchers used hippocampal injections of recombinant AAV encoding mutant forms of APP [24], tau protein [50] or presenilin-1 [49]. Injections into the hippocampus of AAV encoding human Aß42 and Aß40 have been used to selectively increase the levels of these proteins in the rat brain [24]. AAV encoding mutant forms of the tau protein were also used for injection into the substantia nigra of adult rats [44] and the brains of neonatal mice [69]. For detailed information about rodent models of AD, we recommend specialized reviews [70,71,72]. 

Recently, a model of AD has been developed on NHP (rhesus monkeys). The authors injected recombinant AAV1 encoding tau protein with two disease-related mutations (P301L/S320F) into the entorhinal cortex of one cerebral hemisphere. AAV-mediated expression induced propagation of the misfolded tau protein that resulted in a neuroinflammatory response and substantial axonal damage. The results of behavioral tests with created model animals are awaited [54]. 

In the area of AD therapy strategies, a large variety of animal models is a challenge for researchers. To overcome this problem, as well as to standardize existing and developing models, a scoring system has recently been developed to evaluate animal models of AD. The higher the score of an animal model, the better that model reproduces the pathology [71]. This will standardize the analysis and help develop more relevant animal models of the disorder.

### 2.2. Parkinson’s Disease

Parkinson’s disease (PD) is a neurodegenerative disease characterized by the loss of substantia nigra dopaminergic neurons. The overall incidence rate of PD is 17 per 100,000 person-years, with a peak incidence between 70 and 79 years of age. The motor symptoms of the disease include bradykinesia, tremors, and postural instability. PD is also manifested by non-motor symptoms, such as dementia, sleep disturbances, and autonomic dysfunctions [73]. PD is caused by a combination of genetic and environmental factors, but monogenic forms associated with at least 13 loci and 9 genes, including *SNCA* and *LRRK2*, are found in 5–10% of patients [74]. 

The neuropathological hallmark of PD is the abnormal aggregation of α-synuclein in the medulla oblongata neurons that spread upward toward the pons, mesencephalon, and higher brain regions. Many animal models have been developed using autosomal dominant mutations in the *SNCA* gene encoding this protein. However, it was found that the spread of mutant α-synuclein in transgenic mice is limited to the nigrostriatal dopaminergic neuronal system [75,76]. This limitation has been overcome by AAV transduction. Injection of AAV6 encoding human α-synuclein into the vagus nerve in the neck of rats resulted in selective expression and accumulation of α-synuclein in medullary neurons. Thus, retrograde AAV6 transport provided caudo-rostral distribution of α-synuclein, repeating patterns of disease progression [55]. Recently, researchers have used AAV-mediated expression to study the interaction of two genetic factors in the pathogenesis of PD. Leucine-rich repeat kinase 2 is encoded by the *LRRK2* gene, whose mutant forms play a crucial role in the development of PD. Authors used AAV6 for packaging genes encoding the mutant form of α-synuclein (A53T) or the CDS encoding C-terminal portion of LRRK2 (ΔLRRK2) with the G2019S mutation. Co-injections of the AAVs into substantia nigra in rats resulted in motor signs and degeneration of dopaminergic neurons. The detected effect was significantly greater than the effect caused by the expression of the α-syn A53T mutant alone [43]. 

More details on creating animal models of PD can be found in the respective reviews [77,78,79,80].

### 2.3. Amyotrophic Lateral Sclerosis

Amyotrophic lateral sclerosis (ALS) is a heterogeneous neurodegenerative disease characterized by the loss of upper and lower motor neurons and leads to paralysis [81]. The disease affects about 3 individuals per 100,000 population, and the average age of onset is 65 years [82]. It is assumed that the manifestation of the disease is due to a combination of genetic and environmental factors. More than 30 genes are associated with a greater risk of developing the disease, while mutations in *SOD1*, *C9orf72*, *FUS*, and *TARDBP* are identified in 70% of familial ALS. Oligogenic inheritance and pleiotropy characteristics of the disease further complicate the disease modeling [81]. 

The neuropathological hallmark of ALS is the abnormal aggregation of the TAR DNA-binding protein 43 (TDP-43) in motor neurons. TDP-43 is involved in the regulation of RNA processing and normally has nuclear localization. Abnormal TDP-43 aggregations in the cytoplasm were found in almost 97% of ALS patients [83]. In a recent study, AAV1 encoding TDP-43 was injected into the cervical spinal cord of cynomolgus monkeys to model ALS. The animals showed abnormal cytoplasmic localization of exogenous TDP-43, and all monkeys developed progressive motor weakness and muscle atrophy. Interestingly, injections of the same virus into the spinal cord of rats did not result in cytoplasmic localization of TDP-43 [23]. Modeling of ALS in rats was greatly facilitated by the comparative study of spinal cord tropism of AAV1, AAV8, AAV9, and AAV10 after intravenous administration. AAV9 demonstrated the highest transduction efficiency of spinal cord in adult animals and CBA-driven TDP-43 expression yielded highly consistent, dose-dependent motor deficits [56]. In the following studies, authors further improved ALS rat models by using the neuron-specific hSyn promoter and the AAV-PHP.B capsid [51]. 

The most common genetic cause of ALS is the expansion of non-coding GGGGCC repeats in the 5′-UTR of the *C9orf72* gene. mRNA transcripts generated from mutant *C9orf72* aggregate and form intracellular foci [84]. To create the mouse model of ALS, AAV9 expressing 10 or 102 hexanucleotide repeats was injected into the cisterna magna of early postnatal mice. Both mice models were characterized by the formation of intranuclear RNA foci in the CNS; however only animals with 102 repeats generated ALS symptoms such as Purkinje cell apoptosis, neuromuscular junction pathology, and display gait and cognitive deficits. No abnormal aggregates of TDP-43 or neurodegeneration were detected in either mouse model [19]. In another study, the authors developed a mouse model that robustly recapitulates all major hallmarks of ALS. Intracerebroventricular injection of rAAV9 encoding 149 repeats resulted in behavioral impairments, neurodegeneration, and the formation of the TDP-43 aggregates [20]. 

### 2.4. Huntington’s Disease

Huntington’s disease (HD) is a dominant neurodegenerative disorder characterized by choreiform movements, cognitive decline, and behavioral impairment [85]. HD typically presents in the age of 35–50 years and has a prevalence of about 10 cases per 100,000 population [86,87]. The primary cause of the disease is an increased number of CAG repeats over 35 in the first exon of the *HTT* gene. As a result, huntingtin protein (Htt) with elongated polyglutamine (polyQ) tract and neurotoxic properties is produced. The mutant Htt forms aggregates in striatal neurons with subsequent degradation of the affected cells [85]. The AAV-mediated HD modeling allows researchers to investigate the pathogenic effect of Htt poly(Q) sequences of various lengths. Given that the full-length Htt CDS (9429 nt) exceeds the packaging capacity of the AAV vector, truncated N-terminal fragments containing CAG repeats are used to create HD models.

Two independent groups have developed mouse models of Huntington’s disease by direct injection of AAV into the striatum. In the first case, the authors packaged the gene of truncated Htt fragment with 100 CAG repeats in AAV8 to create a rapid-onset HD model in adult mice [45]. The second group created an HD model in juvenile mice using AAV-DJ encoding mutant huntingtin carrying 82 CAG repeats [47]. In both models, the mice developed motor dysfunction and neurodegeneration. Hult and colleagues demonstrated that the expression of mutant huntingtin in the hypothalamus of mice leads to metabolic disturbances. Selective hypothalamic expression of mutant huntingtin resulted in impaired glucose metabolism in injected mice, resembling disease progression in HD patients. In the study, authors used AAV5 encoded N-terminal truncated Htt fragment with 97 CAG repeats [57]. 

Franich and colleagues developed a rapid-onset HD model in adult rats using a chimeric AAV serotype containing equal numbers of AAV serotype 1 and 2 capsid proteins. In the study, AAV-mediated expression of mutant huntingtin with 70 CAG repeats in rat striatum resulted in HD-associated neurodegeneration. The authors noted that injection of this AAV serotype resulted in non-uniform transduction of the striatum with increased transduction of cholinergic interneurons. Moreover, striatal injection resulted in transduction and neuronal death in the globus pallidus and substantia nigra [59]. Two studies have been published on the injection of AAVs into the striatum of neonatal mice for HD modeling. In the first study, the authors used AAV5 to package the gene encoding of Htt fragment with 79 CAG repeats. Striatal injection resulted in the formation of Htt inclusions predominantly in the cerebral cortex and only to a small extent in the striatum. In addition, the formation of intracellular Htt inclusions did not lead to neuronal death or motor dysfunction in animals [46]. However, in another study, striatal AAV2 injections resulted in Htt aggregation at the injection site. The developed model also reproduced motor disturbances, and histological analysis revealed activation of microglia and astrocytes. The study used Htt fragment with 82 CAG repeats [48]. 

In a parallel study in mice and non-human primates, AAV6 was used as a vector for the mutant huntingtin delivery. The recombinant AAV encoding N-fragment of Htt with 103 CAG repeats was introduced into the cerebral cortex. Despite little spreading of the aggregates in both species, the injections induced motor deficits. In mice, these effects disappeared after three months, while in non-human primates, they persisted for more than seven months [16]. 

### 2.5. Spinocerebellar Ataxias

Spinocerebellar ataxia (SCA) is a group of genetically heterogeneous disorders with autosomal dominant inheritance that are characterized by progressive neurodegeneration in the cerebellum (Purkinje cells) and sometimes in the spinal cord [88]. The prevalence of SCA is 3 per 100,000 population and common symptoms are uncoordinated movements including problems with gross and fine motor skills [89].

The most common types of SCAs are repeat-expansion diseases [18]. Curiously, these types of SCAs have not been modeled using a repeat delivery approach, such as described above for Huntington’s disease. Instead, modeling was performed by modulating the expression of the downstream effectors. In particular, SCA type 1 and type 3 are caused by abnormal expansion of CAG repeats in the *ATXN1* and *ATXN3* genes, respectively [90]. Proteins encoded by mutant genes contain an expanded polyQ tract and disrupt many molecular pathways. Among others, a significant decrease in expression of RORα (retinoid-related orphan receptor α) was found in transgenic mouse models. RORα is a transcription factor abundantly expressed in Purkinje cells that plays a role in cerebellum development [91]. In a recent study, the rAAV was used in a mouse model to investigate the contribution of RORα downregulation on cerebellar motor function and the ataxia phenotype. Following intravenous infusion of rAAV-PHP.B, RORα was downregulated in the cerebellum by miRNA expressed using a Purkinje cell-specific L7-6 promoter. RORα silencing caused the degradation of Purkinje cells and the development of the cerebellar ataxia phenotype, confirming the key role of RORα in the etiology of SCA1 and SCA3 [37]. 

SCA type 6 is caused by the polyQ tract expansion in the *CACNA1A* gene, which encodes the α1A subunit of the Cav2.1 voltage-gated calcium channel [90]. A knock-in mouse model that expressed Ca(v)2.1 with 28 polyQ repeats failed to reproduce the SCA6 phenotype [92]. Therefore, the causative role of this channel malfunctioning in the pathogenesis of SCA6 was questioned. The light on the mechanism of SCA6 was shed after the discovery that *CACNA1A* is a bicistronic gene [93]. Due to the presence of a cryptic internal ribosomal entry site (IRES), transcription factor a1ACT is produced from the same mRNA. This factor regulates neural and Purkinje cell development. An AAV gene delivery system was used for SCA6 modeling by overexpressing IRES-driven α1ACT with 33 CAG repeats [17]. Following direct injection of AAV9 into the lateral ventricle of neonatal mice, mutant α1ACT expression was observed predominantly in the cerebral cortex and cerebellum. Injected mice developed pathological features that resemble SCA6 (early-onset ataxia, motor deficits, and Purkinje cell degeneration), thus confirming the causative role of CAG repeats’ expansion in the second cistron of the *CACNA1A* gene.

Mutations in ion channel genes also lead to the development of other types of cerebellar ataxia. Recently, a de novo G354S mutation in the *KCNMA1* gene was identified in a patient with progressive cerebellar degeneration and ataxia [22]. *KCNMA1* encodes the α-subunit of the large conductance calcium-sensitive potassium channel, which is a critical regulator of neuronal excitability. To test the direct impact of G354S on cerebellar function, the mutant *KCNMA1* was delivered into the cerebellum of healthy mice following AAV9 injection. Treated mice developed ataxia, indicating a causative role of the mutant channel in the disease onset and the dominant-negative effect of the G354S mutation. The authors supplemented these results with cellular studies and concluded that the *KCNMA1* G354S variant features a combined loss-of-function with a dominant-negative manifestation.

### 2.6. Spinal Muscular Atrophy (SMA)

Spinal muscular atrophy (SMA) is a neuromuscular disease characterized by progressive loss of motor neurons in the anterior horns of the spinal cord. It is one of the most common genetic causes of infant mortality, with a frequency of 1 in 11,000 births. This autosomal recessive disease results from a deletion or point mutation in the *SMN1* (survival motor neuron) gene leading to a deficiency of the protein product. In humans, low levels of the SMN protein are produced by a centromeric copy of the *SMN2* gene, which is absent in other animal species. The complete absence of the SMN protein is lethal, making it difficult to create animal models with the knockout of a single copy of the gene [3].

Duque and colleagues developed the first model of SMA in large animals, while retaining the baseline expression level of *Smn1*, thus avoiding lethality. Five-day-old piglets were injected with scAAV9 in the cisterna magna. Used vectors encoded an shRNA under the H1 promoter, targeting the transcript of the porcine *Smn1* gene. Knockdown resulted in a 30% reduction in SMN protein in lumbar spinal cord lysates compared to the no-injection control group. The animals developed progressive muscle weakness three to four weeks after injection. Histological staining revealed a significant loss of motor neurons of up to 74% compared to control animals. In addition to pathological changes, the researchers found violations of electrophysiological parameters. Thus, the model reproduced the pathologies observed in patients with SMA [31]. 

### 2.7. Lysosomal Storage Diseases

Lysosomal storage diseases are a group of inherited metabolic disorders characterized by abnormal accumulations of macromolecules in the cell resulting from a deficiency of lysosomal digestive enzymes [2]. These disorders can affect various organs and often impair the function of the central nervous system. Knockout or suppression of the genes encoding the lysosomal enzymes is a convenient method for modeling these diseases in laboratory animals. 

Gaucher disease (GD) is one of the most common lysosomal storage diseases affecting between 1/40,000 and 1/60,000 people in the general population. This autosomal recessive genetic disorder is caused by mutations to the *GBA* gene encoding β-glucocerebrosidase. The mutant β-glucocerebrosidase significantly loses the enzymatic activity, which causes the accumulation of its substrate glucosylceramide in the lysosomes of macrophages. There are three clinical forms of GD: non-neuronopathic (type 1), acute neuronopathic (type 2), and subacute neuronopathic (types 3) [94]. 

Mouse models were developed and used preclinically for type 1 and type 2 of GD, but for a long time there was no optimal model for type 3 [95,96]. A model reflecting GD type 3 was developed in 2019 based on a non-neuronopathic type 1 mouse model Gba^D409V/D409V^. The authors used an intracerebroventricular injection of rAAV1 encoding *GBA*-targeting miRNA for downregulation of the gene expression. The resulting model demonstrated the phenotypes of GD type 3, including neuroinflammation, impaired liposomal metabolism, and progressive neurological aberrations [32]. 

### 2.8. Epilepsy

Epilepsy is one of the most common disabling neurological disorders, characterized by a predisposition to generate recurrent spontaneous seizures [97]. More than half of epilepsies are due to a genetic cause and the use of high-throughput sequencing makes it possible to identify an increasing number of mutations associated with epilepsy [98]. Genetically determined epilepsies are highly heterogeneous and the epilepsy-associated genes can be divided into three groups: epilepsy genes, neurodevelopment-associated epilepsy genes, and epilepsy-related genes. Epilepsy genes cause epilepsy syndromes with epileptic seizures as the main symptom. The second type, neurodevelopment-associated epilepsy genes, are the genes associated with developmental brain malformations that lead to the occurrence of epileptic seizures. The last type, epilepsy-related genes, are the genes associated with any, including systemic, abnormalities, in which one of the accompanying symptoms may be epileptic seizures [99]. 

One of the epilepsy-associated genes is *GNAO1*. The gene encodes the alpha subunit of the G protein (Gαo1) and belongs to the group of epilepsy genes [99]. De novo mutations in this gene cause a severe orphan disorder, GNAO1-encephalopathy. The disease is characterized by early infantile epilepsy, developmental delay, and motor impairment. Twenty-six clinically significant variants in the gene have been described and the phenotypic heterogeneity of the disorder indicates a different mechanism of pathology depending on the specific mutation [100,101]. In a recent article, the authors used the AAV9-mediated expression of mutant Gαo1 proteins to determine pathological mechanisms associated with the two most common *GNAO1* mutations G203R and R209C. Cre-dependent AAVs, encoding mutant or wild-type Gαo1, were injected into the dorsal striatum of adult Drd1Cre and Drd2Cre mice. The use of cre-dependent induction provided a highly specific expression of exogenous Gαo1 in Drd1-expressing direct-pathway neurons (dMSNs) or Drd2-expressing indirect-pathway neurons (iMSNs). Overexpression of both mutant proteins in either MSN subpopulation of the mouse striatum resulted in profound movement control deficits. Based on these results and in vitro experiments described in the paper, the authors concluded that both G203R and R209C mutations in *GNAO1* are characterized by a combination of loss-of-function and dominant-negative manifestations [21]. 

Spontaneous seizures can be caused by a focal disturbance of adenosine homeostasis in the neocortex [26]. Adenosine kinase (ADK) is the main enzyme for the clearance of endogenous anticonvulsant adenosine in the brain [102]. *ADK* is classified as an epilepsy-related gene [99]. AAV8-dependent overexpression of ADK in the murine brain driven by an astrocyte-specific gfaABC1D promoter resulted in focal disruption of adenosine homeostasis and led to neocortical focus of hyperexcitability. This pathological process was also accompanied by a reduced vascular density [26]. Theofilas and colleagues used a similar approach to investigate the role of ADK in the pathogenesis of epilepsy. In their study on mice, an rAAV vector encoding ADK under the same promoter was injected into the CA3 region of the hippocampus. Overexpression of ADK in hippocampal astrocytes caused spontaneous recurrent seizures in the absence of any other epileptogenic triggers [25]. 

To sum up, diverse mutations in a great number of genes are associated with the development of epilepsy. AAV-based gene delivery enables creating animal models with individual clinical variants, which makes it possible to study details of the pathological mechanisms and optimize individual therapy regimens.

### 2.9. Major Depressive Disorder

Major depressive disorder (MDD) is a mental disorder that affects approximately 4% of the world’s population [103]. The symptoms of MDD include depressed mood, anhedonia, and increased apathy [104]. The underlying mechanisms and pathogenesis of depressive disorder are not well established, but studies show that changes in the expression of several genes within different brain compartments can mediate depressive phenotypes [105,106]. 

Studies have found involvement of the basolateral amygdala (BLA) in the pathophysiology of depression [107,108]. The amygdala is an area of the brain that plays a key role in the formation of emotions [109]. AAV-mediated gene expression in mouse models has confirmed several molecular mechanisms of this process [28,61]. A genome-wide association study revealed an association between *SIRT1* expression, a product of which is widely present in BLA, and the risk of major depressive disorder [110,111]. Guo and colleagues used AAV9-mediated expression to test the role of the abnormal expression of *SIRT1* in BLA. Overexpression of the protein under CaMKII (calcium/calmodulin-dependent protein kinases type II) promoter induced depressive behavior in a mouse model, whereas *SIRT1* knockdown with shRNA prevented the formation of a depressive phenotype in stressed animals [28]. Calcineurin (calcium-dependent phosphatase) is another enzyme highly expressed in the amygdala, and its reduced activity contributes to depression symptoms. To reproduce this condition in mice, shRNA was designed to target the *Ppp3ra* gene transcript encoding the regulatory subunit of calcineurin. Following local intraparenchymal injection of AAV2, the shRNA-mediated 35% reduction of *Ppp3ra* transcript in the amygdala was detected. Downregulation of calcineurin in the mouse brain increased the indicators of anxiety behavior and led to the formation of depressive-like behavior [61]. 

The hippocampus is intimately connected with the amygdala and has a role in the pathophysiology of MDD [112]. Several approaches related to the regulation of transcription in this area were implemented for MDD modeling. Ni and colleagues used injections of AAV9 in the hippocampus to study the role of CRTC1 (CREB regulated transcription coactivator 1 protein) in depression. shRNA-mediated suppression of *CRTC1* in the hippocampus directly resulted in depression-like behavior. At the same time, overexpression of *CRTC1* prevented lipopolysaccharide-induced depressive phenotypes in a mouse model [60]. The role of the transcriptional repressors GATA-1 and GATA-2 in the formation of depressive behavior was confirmed using AAV-mediated expression in rat models. Hippocampal expression of these proteins resulted in depressive phenotypes in the forced swim test and the learned helplessness model. Anhedonia caused by the chronic unpredictable stress was blocked by infusion of rats with rAAV-encoded shRNA targeting the GATA-1 transcript [62,64]. 

Some recent clinical and preclinical evidence suggests that neuroinflammation is one of the critical factors in the etiology of depression [113,114]. Inhibition of CaMKII has previously been found to attenuate inflammation-associated protein kinases and mediators of cerebrovascular inflammation in rats [115]. Some molecular mechanisms of neuroinflammation associated with depressive-like behavior have been revealed in rat models with AAV-mediated expression. Hippocampal overexpression of βCaMKII effectively induced the depressive phenotype; by contrast, knockdown of βCaMKII reversed inflammation-related biochemical parameters and significantly relieved core symptoms of depression [63]. 

Another possible cause of depression is disbalance in neurotrophins, growth factors that regulate the development and maintenance of the nervous system. Lin and colleagues [27] studied the peripherally expressed precursor of brain-derived neurotrophic factor (proBDNF) as a pathogenic factor of depression. To simulate peripheral expression, AAV8-proBDNF was injected into adult mice’s skeletal muscles, which enabled a high level of transgene production and secretion. From the bloodstream, proBDNF presumably entered the brain, bypassing the blood–brain barrier. A total of 4–8 weeks post-injection, treated animals developed depressive behavior: increased immobility time in the tail suspension test and forced swim test, reduced sucrose consumption, and decreased grooming time after sucrose spraying. Treatment with proBDNF neutralizing antibodies alleviated depressive-like symptoms, thus confirming the hypothesis that peripheral proBDNF is a primary trigger for depression. 

Major depressive disorder is one of the most complex diseases, which is apparently significantly associated with an imbalance in gene expression. Some genetic factors have been identified, while others await research. AAV-mediated expression can be used to test new candidate genes as well as to study the interaction of several risk factors in one model.

## 3. Conclusions and Perspectives

Using AAV for modeling human CNS diseases in laboratory animals clarified the role of the disease-causing genes and the involvement of the specific pathways and shed light on the pathogenesis of disorders such as Alzheimer’s disease, Parkinson’s disease, Huntington’s disease, ataxias, amyotrophic lateral sclerosis, epilepsies, and depression. Recombinant AAV vectors are convenient due to their flexible design and allow manipulation of the transgenes, promoters, additional regulatory elements, and reporter genes. A variety of neurotropic serotypes and established routes of the virus administration allow targeting different areas of the CNS according to the characteristics of the disease. Of particular value are models of the neurological disorders created using AAV in large animals with highly organized nervous systems such as NHP. The AAV platform also has several bottlenecks, including limited packaging capacity of the AAV particle, restricted spreading of the virus across brain structures, and low transduction efficiency of the individual types of neurons. Current studies in the field aim to overcome these limitations and expand the scope of AAV for modeling a greater variety of diseases.

Strategies to overcome the limited packaging capacity of AAV vectors (which cannot hold more than ~5 kb of DNA) can be borrowed from the rapidly developing field of gene therapy. For the delivery of large therapeutic transgenes, dual AAV vector systems are employed [116]. The over-sized transgene is divided between two discrete AAV vectors. After simultaneous transduction of cells, segments of the AAV genomes undergo homologous recombination or trans-splicing and recover the full-length coding sequence. It is promising to use similar dual vector systems for modeling neurological disorders that require delivery of large transgenes. 

One of the hot research topics is the creation of AAV capsids with new and/or improved characteristics by directed evolution and rational design [13,117,118]. For example, a recent high-throughput screening identified two synthetic AAV serotypes with high neural stem cell transduction efficiency in the ventricular–subventricular zone of mice brains. In the study, authors screened 177 variants of AAV capsids in experiments with intraventricular injections. The two identified synthetic serotypes are the peptide-modified derivative of wild-type AAV9 and the peptide-modified derivative of wild-type AAV1 [119]. 

In addition, researchers are improving technologies for analyzing the distribution and efficiency of the developed AAVs. Westhaus and colleagues have developed a kit for high-throughput screening of AAV variants based on the barcoding systems for the next-generation sequencing (NGS). The approach allows analyzing the distribution of AAVs in tissues both at the DNA level and at the RNA/cDNA level. Using the described technology, individual researchers can generate identical kits customized for their projects [120]. In Smith’s paper, the authors developed a methodology for automated screening of AAV serotypes by the fluorescent signal of the reporter gene in brain slice images. The developed scripts automatically divide the brain slice into certain areas and quantify the percentage of transduced neurons in the cortex, striatum, and hippocampus [121]. 

The AAV capsid can also be modified to target cellular compartments. A recently published groundbreaking study reported the development of a mouse model of Leber hereditary optic neuropathy. Authors fused the AAV2 capsid with a mitochondrial target sequence (MTS) and packaged in the resulting MTS-AAV2 the mutant gene of human NADH ubiquinone oxidoreductase subunit 4 (ND4). Injections of the MTS-AAV2 into fertilized mouse oocytes led to the successful delivery of the transgene into mitochondria. The generated mice demonstrated early vision loss, which began at three months, progressing to blindness eight months after birth. Authors found that the genetic construct did not integrate into the host’s mitochondrial DNA and remained episomal [122]. 

The development of novel animal models recapitulating human disorders by means of AAV broadens the spectrum of available platforms for drug screening and testing of advanced therapy medicinal products. Hopefully, this will shorten the time of drug discovery for currently incurable conditions.

## Figures and Tables

**Figure 1 biomedicines-10-01140-f001:**
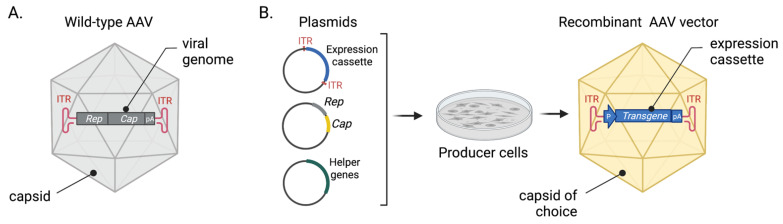
A general principle of creating recombinant AAV vectors. (**A**) Structure of the wild-type AAV. (**B**) System for rAAV production, and basic structure of the recombinant AAV vector.

**Figure 2 biomedicines-10-01140-f002:**
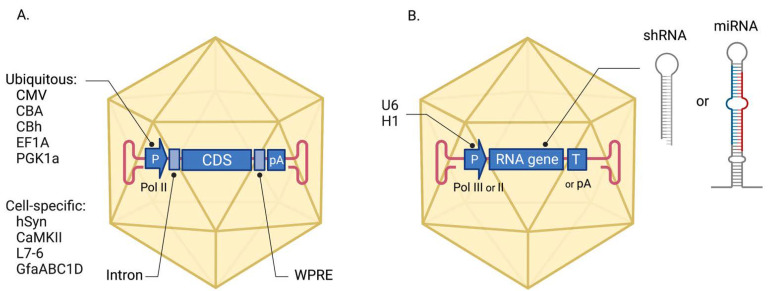
Design of the AAV expression vectors for modeling neurological diseases. (**A**) AAV expression cassette suitable for the transfer of protein-coding transgenes. (**B**) Configuration of the AAV vector utilized for gene suppression via RNAi mechanism. P, promoter for RNA Pol II or III as indicated; CDS, the protein-coding DNA sequence; pA, poly(A) signal for Pol II-driven transcription termination; WPRE, woodchuck hepatitis virus posttranscriptional regulatory element; T, T-stretch for Pol III-driven transcription termination; shRNA, small hairpin RNA; miRNA, artificial microRNA.

**Figure 3 biomedicines-10-01140-f003:**
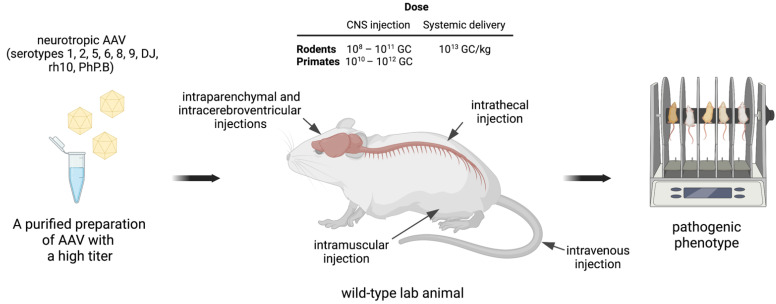
Creating models of human neurological disorders by targeting the CNS of wild-type laboratory animals with AAV. The most critical steps include obtaining custom-designed neurotropic rAAV, virus administration into a laboratory animal at a selected dose and by the optimal route, and observing the phenotype after the disease onset. Neurotropic serotypes of AAV and routes of administration used by researchers to model neurological diseases in rodents and primates are exemplified. The doses of AAV used for CNS injections and systemic delivery are shown. GC/kg, genome copies per kilogram of the body weight.

## Data Availability

Not applicable.

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
