# Peer review of "Adeno-Associated Viruses for Modeling Neurological Diseases in Animals: Achievements and Prospects"

_biomedicines, 2022, doi:10.3390/biomedicines10051140_

Round 1
Reviewer 1 Report
In this manuscript, the usefulness of AAV platform in producing animal models of human neurological disorders (e.g., Alzheimer’s disease, Parkinson’s disease, ataxias, etc.) is described. This review is concisely and readably summarized, and will provide neuroscience researchers with valuable information. The reviewer recommends that the figure concisely illustrating the AAV serotypes including both naturally occurring and gene engineered types is included in the manuscript. The figure makes this review more informative.
Author Response
We appreciate the reviewer for the time spent on a careful reading of our manuscript and overall positive feedback.
Indeed, dozens of naturally occurring and engineered AAV serotypes are described to date. Remarkably, a single amino acid difference in the capsid protein can significantly affect the properties of the serotype such as tissue tropism. Illustrating the variety of AAV serotypes is beyond the scope of this review. Instead, we focused on neurotropic AAV serotypes popular among researchers for modeling neurological disorders in the lab animals. To emphasize this, we edited Figure 3 and its description (page 6, lines 260-266) and included the corresponding sentence in the text (page 6, lines 273-275). For readers interested in this topic, we have referred to excellent reviews discussing neurotropic serotypes and their features.
Reviewer 2 Report
Evgenii Lunev and coworkers excellently reviewed the topic entitled “Adeno-associated viruses for modeling neurological diseases in animals: achievements and prospects” with a particular focus on the AAV-mediated modeling of ultra-rare disorders affecting the central nervous system.
Overall, the authors provide an appreciable review that is worth getting published in the MDPI Biomedicines, however, I recommend a detailed revision addressing the following minor issues carefully:
- Page 2, line 52: Define what is C9orf72-mediated ALS? ïƒ chromosome 9 open reading frame 72 (C9orf72)-mediated ALS
- Page 2, lines 52-54: Please mention the mutation type and gene name that causes the lysosomal storage diseases and SMA in the following sentence of the manuscript Another mutation type results in the deficiency of the functional gene product, as in the case of lysosomal storage diseases [3] or SMA[4].
- Page 6, line 246: Define what is GC/kg? It is better to define after its first appearance in the manuscript ïƒ genome copies/kilogram of the body weight (GC/kg)
- Page 2, line 95: Wild type AAV (Parvoviridae). This statement may give the wrong impression to the non-biology audience that Parvoviridae is an example of Wild type AA Hence, better to rephrase the statement. For example, Wild type AAV is a member of the genus Dependovirus, of the familyParvoviridae,
- No information was provided for FDA approved AAV product Zolgensma for the treatment of SMA type 1in infants. Better to include FDA-approved AAV products for neurological diseases, if any other than Zolgensma.
I recommend the authors include the following perspectives in the conclusions and perspectives section. Don’t hesitate to rephrase the following statements and add any other perspectives.
- Although AAV serotypes showed some sort of tropism to the CNS, liver sinusoidal capture of the AAV vectors hinders the application potential of AAV to reach their target organs/cells (motor neurons in the case of AAV9). This resulted in increasing the viral dose (3×1014 GC/kg) to obtain a therapeutic level of protein expression in the target cells. However, such a high viral dose was unexpectedly associated with severe hepatotoxicity, thus leading to the death of children in a recent clinical trial(1089/hum.2020.182). This clearly demands the urgency of strategies that reduce the viral dose to the liver without compromising the level of therapeutic protein expression at the target organs/cells. In this regard, Kataoka et al. developed a transient and selective stealth coating of liver sinusoids using two-arm-PEG-OligoLysine to inhibit the sinusoidal sequestration of viral vectors to relocate the vectors from the liver sinusoid to their intrinsic targeted organs/cells (10.1126/sciadv.abb8133).
- Please specify the limitations of AAV vectors for gene therapy. For example, the packaging capacity of natural AAV and rAAV is limited to ~ 4.7 kb and 4 kb, causing a major challenge to deliver large-sized gene products. Therefore, genes exceeding ~4 kb represent an oversized cargo for AAV vectors. In particular, packaging Streptococcus pyogenes (SpCas9) and a gRNA together (~ 2 kb) and laminin α2 cDNA (> 9 kb) precluding their packaging into a single AAV vector, thus limiting the application potential of AAVs. In such a scenario, AAV-inspired vectors such as polyplex micelle packaging single-stranded DNA would be advantageous to package large-sized gene products (10.1021/acsnano.9b04676)
Author Response
We thank the reviewer for the time spent on a careful reading of our manuscript, positive feedback and valuable recommendations.
1. Following the comment, we removed the abbreviation of the gene C9orf72 from the introduction (page 2, line 49) and defined C9orf72 as chromosome 9 open reading frame 72 in the subchapter “AAV expression cassette” where mentioning C9orf72 is more relevant (page 4, lines 191-192).
2. In the introduction, we mention the classification of the disease-causing mutations based on the effect on the gene function. We rewrote part of the text concerning mutation types and associated disorders. We hope the new version of the text is clear without using the specific terms for the mutation types (page 2 lines 49-58). Lysosomal storage diseases are large group of disorders caused by mutations in different genes. Therefore, naming a particular gene responsible for these disorders is not possible. To emphasize the diversity of these pathological phenotypes, we changed the wording to "a group of lysosomal storage diseases." (page 2 line 58). In the section devoted to lysosomal storage disease, we added citation of an excellent review by Platt et al. (2018) that lists causative genes (page 14 line 574). The causative gene for the SMA is detailed in the section discussing modeling of the SMA pathology (page 13 lines 551-556).
3. Following the remark, we have defined GC/kg at the first mention (page 7 line 320).
4. The text was changed according to the recommendations (page 3 line 124).
5. Following the recommendation, we mentioned in the introduction the first FDA-approved AAV-based gene therapy for inherited disorders of the central nervous system: Zolgensma for SMA and Luxturna for Leber congenital amaurosis (page 2, lines 86-88). We included references that provide detailed information on these AAV-based drugs as well as the development of gene therapy for other neurological disorders.
6. It is a fair critique that the review does not address the issue of AAV toxicity and in particular hepatoxicity. To fill this gap, we added a discussion of the AAV toxicity following the CNS injections and systemic administration to the section “Targeting neuronal tissues” (page 7, lines 322-327). The technology of selective stealth coating of liver sinusoids using two-arm-PEG-OligoLysine is indeed very promising for reducing effective virus dose and managing the AAV hepatotoxicty. However, this technology is less relevant to the field of creating animal models using AAV as compared to gene therapy studies. In the disease modeling, the effects caused by the administration of a high dose of AAV can be controlled by treating a control group of animals with AAV encoding the reporter gene. And since the issue of AAV toxicity is very broad, we reasoned it more appropriate to cite two recent reviews devoted to AAV toxicity and ways of its management in regard to gene therapy for neurological disorders (Perez et al. 2020, Hudry et al. 2019).
7. As noted, the limited capacity of AAV poses a significant challenge for the delivery of large genetic constructs. While non-viral vectors including polyplex micelles hold a great promise for packaging and delivering large-sized genes, such technologies are beyond the scope of our review. However, AAV-based strategies currently being developed for gene therapy would be useful for transferring large genes during disease modeling. Dual-vector systems divide oversized genes between two constructs. Upon delivery, these sequences allow the reconstruction of the full-length coding sequence by homologous recombination or trans-splicing (McClementsa and MacLarena, 2017). To the best of our knowledge, this method has never been used to model diseases so far. We included a brief description of this technology as a potential strategy to overcome AAV limited packaging capacity (pages 16-17, lines 725-742)